# Robust medical image segmentation by adapting neural networks for each test image

## Abstract

Performance of convolutional neural networks (CNNs) used for medical image analyses degrades markedly when training and test images differ in terms of their acquisition details, such as the scanner model or the protocol. We tackle this issue for the task of image segmentation by adapting a CNN ($C$) for each test image. Specifically, we design $C$ as a concatenation of a shallow normalization CNN ($N$), followed by a deep CNN ($S$) that segments the normalized image. At test time, we adapt $N$ for each test image, guided by an implicit prior on the predicted labels, which is modelled using an independently trained denoising autoencoder ($D$). The method is validated on multi-center MRI datasets of 3 anatomies. This article is a short version of the journal paper (Karani et al., 2021).

**Keywords:** medical image segmentation, cross-scanner robustness, domain generalization.

## 1. Introduction

CNNs excel at function approximation within the probability distribution of the training dataset, but make unreliable predictions for out-of-distribution inputs. Changes in the input distribution (domain shifts) are common in medical imaging due to acquisition-related variations such as those in imaging protocols, scanning parameters as well as inherent hardware differences in different scanners. Accordingly, the performance of a CNN trained using a dataset obtained from one hospital typically degrades substantially when tested on images in another hospital. Arguably, such lack of robustness is one of the main factors hindering large-scale clinical adoption of CNN-based methods.

In the literature, the domain shift problem is tackled in several machine learning settings (Table 1). Among the first 4 settings, domain generalization (DG) is the most attractive as it leverages labelled datasets to learn robust mappings that can be directly used for prediction in unseen domains. Although DG improves robustness, there still remains a gap to the benchmark performance. On the end of the spectrum, unsupervised learning methods (Van Leemput et al., 1999) work robustly across acquisition-related variations, but typically rely on deformable registration and are restricted to neuroimaging data. Test-time adaptation (TTA) methods (He et al., 2020; Sun et al., 2020; Karani et al., 2021) combine the advantages of both settings by first leveraging the power of CNNs to learn from labelled datasets and further fine-tuning them specifically for each test image.

## 2. Method

We propose TTA for robust image segmentation. The two main questions in TTA are 1) which parameters to adapt for each test image? and 2) how to drive the TTA? Noting that acquisition-related domain shifts manifest primarily as contrast changes, we design

| Setting | Source Domain (SD) | | Target Domain (TD) | |
|---|---|---|---|---|
| | Data | Algorithm | Data | Algorithm |
| New CNN | $\{x_{SD}, y_{SD}\}$ | $min_\theta L^{Seg}_{SD}$ | $\{x_{TD}, y_{TD}\}$ | $min_\theta L^{Seg}_{TD}$ |
| TL | $\{x_{SD}, y_{SD}\}$ | $min_\theta L^{Seg}_{SD}$ | $\{x_{TD}, y_{TD}\}$ (few) | Init. at $\theta^*_{SD}$, $min_\theta L^{Seg}_{TD}$ |
| UDA | - | - | $\{x_{SD}, y_{SD}, x_{TD}\}$ | $min_\theta L^{Seg}_{SD} + L^{Inv}_{SD,TD}$ |
| DG | $\{x_{SD}, y_{SD}\}$ | $min_\theta L^{Seg}_{SD} + L^{Inv}_{SD}$ | $x_{TI}$ | $\hat{y} = S_{\theta^*_{SD}}(x_{TI})$ |
| DG with TTA | $\{x_{SD}, y_{SD}\}$ | $min_\theta L^{Seg}_{SD} + L^{Inv}_{SD}$ | $x_{TI}$ | Init. at $\theta^*_{SD}$, $min_\theta L^{TTA}_{TI}$ |
| Unsupervised | - | - | $x_{TI}$ | $min_\theta P(S_\theta(x_{TI})) \, P(x_{TI}\|S_\theta(x_{TI}))$ |

Table 1: Machine learning settings for dealing with domain shifts. $TI$ refers to a single test image. 'New CNN' refers to separate learning in each domain, TL to transfer learning, UDA to unsupervised domain adaptation, DG to domain generalization and TTA to test-time adaptation. $L^{Seg}$, $L^{Inv}$ and $L^{TTA}$ stand for a supervised segmentation loss, a feature invariance loss (across SD and TD or across different SDs), and a test-time adaptation loss, respectively. One of the main challenges in TTA is the formulation of $L^{TTA}$ - how to drive TTA in the absence of any labelled samples of the target domain.

the segmentation CNN as a concatenation of a relatively shallow normalization CNN ($N_\phi$), followed by a deep CNN ($S_\theta$) that segments the normalized image. We train both $N_\phi$ and $S_\theta$ on the SD. Then, we fix $S_{\theta^*_{SD}}$ and adapt $N_\phi$ for each test image. We drive the TTA by requiring that the predicted labels are *plausible*, as gauged by a denoising autoencoder (DAE), $D_{\psi^*_{SD}}$, trained on source domain labels. DAEs can leverage long-range spatial correlations and shape cues to suggest corrections in the predicted labels. Specifically, we carry out the following optimization for each test image.

$$min_\phi \, L^{Seg}\left(y_c, \, D_{\psi^*_{SD}}(y_c)\right), \, y_c = S_{\theta^*_{SD}}(N_\phi(x_{TI}))$$

## 3. Experiments and Results

We validate the proposed method on multi-center MRI datasets of 3 anatomies: brain, prostate and heart. A summary of the main observations (Table 2): 1) DG methods based on data augmentation and meta learning substantially improve robustness, but there still remains a gap to training separately on each TD, 2) the proposed TTA can successfully bridge a large portion of this gap, and in some cases, even provides better results than the benchmark, 3) TTA can achieve comparable performance to UDA methods, while not requiring SD images and labels to be available in the TD, 4) analysis experiments validate the hypothesis that fine-tuning all CNN parameters for each test image hurts performance. Please refer to (Karani et al., 2021) for further details.

## 4. Conclusion

We proposed test-time adaptation for robust medical image segmentation. Validation with multiple datasets and anatomies showed the promise and generality of the method over approaches such as data augmentation, meta learning and unsupervised domain adaptation.

| Anatomy | | Brain | | | Prostate | | |
|---|---|---|---|---|---|---|---|
| Train (down) \| Test (right) | | SD | $TD_1$ | $TD_2$ | SD | $TD_1$ | $TD_2$ |
| SD (Baseline) | | 0.853 | 0.588 | 0.107 | 0.840 | 0.586 | 0.609 |
| $TD_n$ (Benchmark) | | - | 0.896 | 0.867 | - | 0.817 | 0.834 |
| DG (Meta Learning) (Dou et al., 2019) | | 0.870 | 0.693 | 0.073 | 0.913 | 0.751 | 0.781 |
| DG (Data Aug. (DA)) (Zhang et al., 2020) (Strong baseline) | | 0.876 | 0.753 | 0.083 | 0.911 | 0.769 | 0.786 |
| SD + DA + TTA (Adapt $\phi$, using DAE) (Proposed) | | - | **0.800** | 0.733 | - | 0.790 | **0.858** |
| SD + DA + TTA (Adapt $\phi, \theta$, using DAE) | | - | 0.671 | 0.650 | - | 0.718 | 0.606 |
| UDA (Invariant features) (Kamnitsas et al., 2017) | | - | 0.798 | 0.083 | - | **0.793** | 0.802 |
| UDA (Image-to-Image translation) (Huo et al., 2018) | | - | 0.639 | **0.813** | - | 0.694 | 0.747 |

Table 2: Mean Dice scores over 3 runs. See (Karani et al., 2021) for dataset details.

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
