# OpenReview forum: "Robust medical image segmentation by adapting neural networks for each test image"
_MIDL.io/2021/Conference/Short — MIDL 2021 Poster_

### Official Review · Reviewer_T4JJ · 2021-04-20

**Confidence:** 5
**Final Rating:** 4

**Summary:**

This paper is a short version of a recently published journal paper. It proposes an approach for robust medical images segmentation under domain shift. The method adapts neural networks trained in the source domain to each test image in the target domain, by utilizing a denoising autoencoder. Extensive evaluation on multiple datasets and anatomies demonstrate the effectiveness of the method.

**Strengths:**

- Performing adaptation on each test image is a more practical setting, which does not require the availability of source image during adaptation process or a pre-collected dataset of target images. The proposed method is well-motivated for the test-time adaptation.

- Experimental evaluation is extensive and sufficiently demonstrate the effectiveness of the method.

**Weaknesses:**

- The method requires to train a denoising autoencoder during the source domain training, which is not a common process in neural network training. In this regard, the proposed method cannot be applied to a "off-the-shelf" model, which has already been trained in the source domain and the training a denoising autoencoder is not feasible anymore.

- The performance of this method largely depends on the quality of the denoising autoencoder, but training the denoising autoencoder seems to be non-trivial task, which requires careful design to create noising segmentations. It is not sure whether such design needs to be specifically tuned for each dataset/task.

**Deanonymize Review:**

no

**Justification Of The Rating:**

This paper presents a well-motivated method for an interesting problem and would have good interest to the conference attendees.  The evaluation is also extensive and clearly validates the proposed method.

**Paper Type:**

both

**Special Issue:**

no

---

### Official Review · Reviewer_4aLS · 2021-05-07

**Confidence:** 4
**Final Rating:** 4

**Summary:**

The authors introduce test-time adaptation for medical image segmentation of MRI signal to segment the brain, prostate and heart. They achieve this by introducing a shallow network to normalize the input image before the (fixed) segmentation network. For each test image, the normalization network is driven to produce an output image that creates reasonable segmentations, based on the segmentations produced on the training dataset. The method is compared against other domain shift solutions.

**Strengths:**

The paper showcases another area where TTAs prove to be powerful. The paper has a strong evaluation, and the authors show a good understanding of the other available methods for domain shift solutions. The short paper is also easy to follow, well-written.

**Weaknesses:**

Keep in mind, that the following weaknesses are all handled in the full version of this paper, which has been already published, so my comments are not for the conducted study but the form of the paper presented here.

Experiments on heart scans is mentioned, yet never evaluated.
As the paper notes that "acquisition-related domain shifts manifest primarily as contrast changes". As the normalization is the focal step of the presented method, the effects of this normalization would be interesting to see visually, or at least discussed. This is done somewhat in the full paper.

**Deanonymize Review:**

yes

**Justification Of The Rating:**

Comparing against strong other methods proves the effectiveness of TTA for medical image segmentation. The authors are thorough both in presenting the related works and in evaluating their method.

The three page format does not do justice to the work presented here, although the paper is very concise and compact (both table 1 and 2 are slightly outside the margins of the page) some important information is missing from the paper.

However the project is well-founded and the results are impressive.

**Paper Type:**

methodological development

**Special Issue:**

no

---

### Meta-Review · Area_Chair_U5S7 · 2021-05-07

**Recommendation:** Accept (Poster)
**Confidence:** 5

**Metareview:**

Both reviewers highlight that this is a strong paper and should be accepted.

---

### Decision · Program_Chairs · 2021-05-11

Accept (Poster)